# Integrating, Indexing and Querying the Tangible and Intangible Cultural Heritage Available Online: The QueryLab Portal

**Maria Teresa Artese *** and **Isabella Gagliardi**

IMATI-CNR (National Research Council), Via Bassini, 15, 20133 Milan, Italy; gagliardi@mi.imati.cnr.it
* Correspondence: artese@mi.imati.cnr.it

**Abstract:** Cultural heritage inventories have been created to collect and preserve the culture and to allow the participation of stakeholders and communities, promoting and disseminating their knowledges. There are two types of inventories: those who give data access via web services or open data, and others which are closed to external access and can be visited only through dedicated web sites, generating data silo problems. The integration of data harvested from different archives enables to compare the cultures and traditions of places from opposite sides of the world, showing how people have more in common than expected. The purpose of the developed portal is to provide query tools managing the web services provided by cultural heritage databases in a transparent way, allowing the user to make a single query and obtain results from all inventories considered at the same time. Moreover, with the introduction of the ICH-Light model, specifically studied for the mapping of intangible heritage, data from inventories of this domain can also be harvested, indexed and integrated into the portal, allowing the creation of an environment dedicated to intangible data where traditions, knowledges, rituals and festive events can be found and searched all together.

**Keywords:** inventory integration; data visualization; common metadata model; ICH ontology; semantic query expansion; word cloud; intangible heritage; living heritage

## 1. Introduction

Cultural mapping is crucial for the safeguarding of the world's tangible, intangible, and natural heritage, as UNESCO stated in 2003. Since then, several countries have created lists or inventories to collect and preserve their culture, allowing the participation of stakeholders and communities to protect and preserve intangible assets, promoting and disseminating their knowledge.

Moreover, by the integration of data harvested from different archives, it is possible to compare cultures and traditions, creating connections and showing how people from different places on earth have in common more than expected.

The QueryLab portal originates from the need to simultaneously query different archives available both locally and online, to facilitate and speed the process of searching data regardless of the language in which the content is expressed. To facilitate the search for different types of users, which in the field of cultural heritage ranges from the merely curious to experts in the field, from people who have no knowledge of the web to experienced geeks, and from students to more mature ladies and gentlemen, different search modes have been studied and implemented for their different information needs. So guided tours, query expansion and semantic tools also have to help the user to access large databases simultaneously with a single click, without being lost in front of a void query field to fill in. In addition, a specific section dedicated to intangible heritage has been developed, integrating new methods to show the results, to provide a graphical and interactive overview of the distribution of the items found, as an alternative to the classic mosaic view.

To query data in an integrated way, two different approaches have been defined and followed: on the one hand, the use of web services provided directly by online databases, e.g., Europeana, and on the other hand, the ingestion of local and online databases that do not provide any web services, using a new three-level metadata structure in a querable/searchable data source. This last feature is oriented and focused mainly on the collection and preservation of data related to intangible heritage, which by its nature is more difficult to find and complex to document.

Finally, the paper reports the ongoing study and integration of an ontology capable of producing linked open data for the local heritage, to allow the linking of the database created with the data cloud and prevent the problem of data silos. In this paper, we present the study of the ontology at the abstract entity level without addressing the requirements in the metadata schema and implementation levels, which will be approached and refined in future studies.

The contribution of this work is to offer an overview of QueryLab's functionalities, an overall and in-depth analysis of the pipeline that starts from the identification of archives on cultural assets, preferably intangible, the steps necessary for their integration if provided with web services or their storage using the newly defined data model, in view of their exposure on the web as LOD. The last steps concern content extraction and the definition of query and visualization tools available for different categories of users, presented through the different activities, choices and difficulties encountered.

The work is structured as follows: Section 2 quickly outlines related work, then in Section 3, our approach for integrating tangible and intangible cultural heritage data is described in detail. Section 4 presents the definition of a data model specifically designed for intangible assets, and Section 5, the proposal of an ontology on the data model just presented. An overview of the multimodal search engine is presented in Section 6, and Section 7 provides a discussion of the innovative features of QueryLab. Section 8 concludes the paper with conclusions and future work.

## 2. Related Works

Integrating data from different sources is a topic of great interest, since data are assets that do not become obsolete, unlike querying and visualization tools. Speaking of this, it is essential to mention "Europeana", the European digital cultural platform that includes, gathered in a single platform, information from more than 10 million cultural artifacts presented in different ways and oriented to today's users.

Inventories for intangible assets are less widespread: "Map of e-Inventories of ICH" managed and updated Memória Imaterial represents the archives on the map, along with a short description and web address for each of them [1].

UNESCO itself offers, on its website, query and immersive navigation tools for the individual intangible assets included in its Intangible Cultural Heritage Lists and in the Register of Good Safeguarding Practices.

From the methodological point of view, Siqueira et al. [2] present a systematic review of workflow models for data aggregation in the specific context of cultural heritage. In addition to the identification of the projects, the different steps for aggregation are investigated, along with the technologies used and the (semi-)automatic solutions adopted.

The CARARE project [3] is based on a central repository called the monument repository (MoRe), where all metadata is stored and enriched before being mapped to the Europeana data model (EDM) and provided to Europeana.

In the HOPE project [4], a Software Toolkit is presented as an "ideal candidate for the creation of sustainable, extensible, scalable, and dynamic data aggregation infrastructures for cultural heritage".

The Digital Public Library of America (DPLA) developed a workflow for the ingestion of U.S. libraries, archives and museums metadata, defining an extension of the EDM and, like the EDM, incorporates or references a variety of standards and templates [5].

The ARIADNE project aims to ensure the semantic interoperability of archeological data, using the ARIADNE Catalogue Data Model (ACDM) [6].

The Data Aggregation Lab (DAL) software implements the cultural heritage metadata aggregation workflow, based on Europeana's aggregation solution, currently adopted [7].

Deligiannis et al. [8] present a system for collecting, managing, analyzing and sharing diverse, multi-faceted cultural heritage/tourism-related data to assist scientists in the cultural heritage domain and tourism stakeholders to gather and synthesize scattered information to exploitable knowledge.

The Europeana data model (EDM), which is widely used and is the basis for many extensions, has been developed by the Europeana community with the aim of providing a flexible tool to represent different perspectives on a given cultural object [9].

Alexiev [10] gives an overview of the main ontologies and models used in GLAM (acronym for "galleries, libraries, archives, and museums"), describing among others, EDM and CIDOC-CRM (the International Council of Museums model), to arrive at the conclusion that semantic data integration is the key to interlinking CH data across time and borders.

Among the ontologies, it is impossible not to mention ArCo ("Architettura della COnoscenza" in Italian). It is composed of seven ontologies, structured in the field of cultural heritage, according to the ICCD-MiBAC models. Two ontologies include high-level concepts and generic relationships between modules while the other five are focused on cultural assets and their features [11].

Szekely et al. [12] report the activities and problems faced in the mapping of the Smithsonian American Art Museum in Linked Open Data format. An extension of EDM, called SAAM, has been introduced. It is a significative example of how the database-to-RDF process can be implemented for tangible heritage.

Wijesundara and Sugimoto [13] and Wijesundara et al. [14] propose a new model, called CHDE (cultural heritage in digital environments) because they consider neither EDM nor CIDOC-CRM suitable for intangible heritage. The main feature of the model is instantiation that "[ . . . ] acts as a bridge to aggregate those resources related to intangible cultural heritage." The CHDE model collects in one instance all digital resources related to a specific moment or occasion. The instance is transformed into a digital archive record according to the One-to-One principle of metadata [15,16], which is necessary to distinguish digital copies from their physical source.

Data aggregator sites require a search engine that enables multiple ways to search for different types of data, users, and information needs. The large amount of data resulting from the aggregation process and its heterogeneity raises specific challenges for the research and exploration. According to [17,18], users most appreciate the availability of search tools that put them at the center, through a design that fully enhances the user experience. The different types of users with their information needs require specific and specialized solutions. Pre-packaged queries from experts, query expansions, the use of glossaries or specialized vocabularies, as well as real-time extraction of terms used in the archives provide tools to improve the users' experience of searching and browsing. [19–21].

There is a rich literature on query expansion in the information retrieval (IR) area, starting from the 1960s. In 1971, Rocchio [22] used QE through the "relevance feedback method" in a vector space model.

Recently [23], many researchers are using QE techniques to work on personalized social services [24]. Carpineto and Romano [25] reviewed the main QE techniques, data sources, and features in an information retrieval system [26]. Query expansion methods are also widely used in the web [27,28].

Word clouds are a tool created several years ago for displaying the most frequent terms in a text or article in a cloud, in which the size of individual terms is proportional to their frequency. One of the most popular tools is Wordle from IBM. The ease of use and immediacy in carrying the message have prompted researchers to use it in text mining or the analysis of unstructured text data, defining metrics and approaches to calculating the frequency of each term in the corpus [29–31].

The huge digital cultural heritage on the web requires new ways of display and higher levels of interaction for both academic scholars and web users. Faced with user demands to move beyond standard search-centric and grid-based interfaces, many approaches have been experimented with to enable visual access to cultural collections and the definition of interactive visualizations, designed specifically for the rich collections of the cultural heritage sector. In [32], different cultural heritage visualization interfaces were critically analyzed in relation to data, users, purposes, tasks considering granularity and interactivity, and visualization methods.

While there are several sites that are the entry point for museum or tangible cultural heritage contents, to our best knowledge, QueryLab is the first that also deals with intangible assets.

## 3. Materials and Methods

This paper presents the pipeline for the creation of the QueryLab portal related to tangible and intangible cultural assets, offering tools for users with different backgrounds and capabilities.

The QueryLab platform has been designed to facilitate the approach to different cultural heritages allowing the query of multiple databases at the same time: a single query is performed in a simple and transparent way for the user and results are organized to have an immediate overview of the different types of data retrieved.

The architecture of QueryLab (Figure 1) is composed as follows:

1. Integration of cultural heritage data, both local and through web services, to make the source of information transparent to the user;
2. Definition of a data model designed specifically to intangible assets, for the collection and indexing of information from 'silos' inventories not providing web services;
3. Multimodal search engine, oriented to user experience.

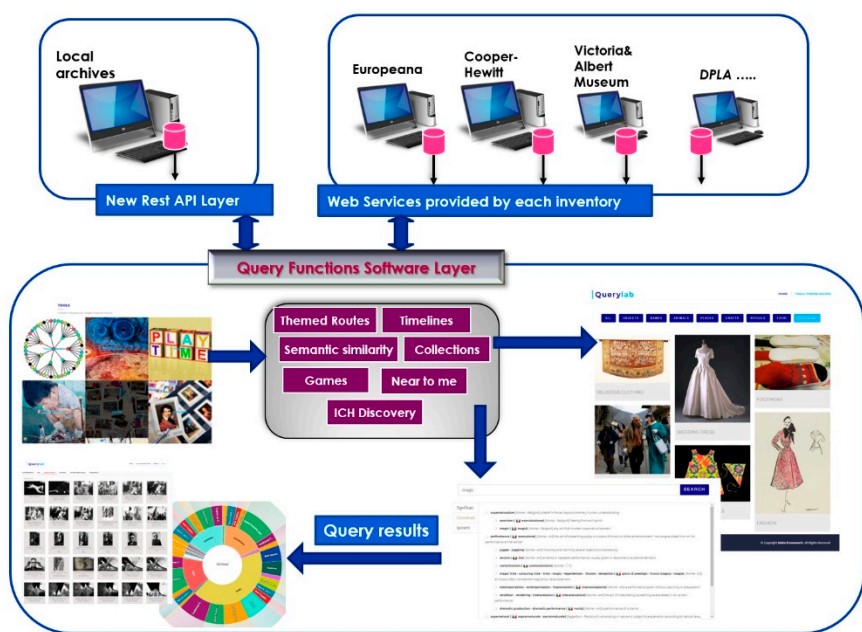

**Figure 1.** Overview of the QueryLab schema.

The first step for QueryLab design is to search and find different websites dedicated to cultural heritage inventories. There are different types of inventories: those who give data access through web services and others which are closed to external access and can be visited only through dedicated web sites, generating data silo problems.

For the first type of inventory, it is possible to develop query tools using the API web services when provided, so we studied and tested different GLAM web services to check

their integration in the portal. For the other type, closed inventories, we studied a model capable of high-level indexing of the stored data, either manually or by implementing specific software modules, to be used once, capable of capturing the basic information and saving it in a web-service ready inventory. Detailed description of the model can be found later.

Focusing on the first type, web services allow the interaction with the underlying database, allowing the extraction of data through a query that is executed through a GET request within the HTTP protocol. Often, the developer of the API service also provides tools and supporting documents for use, sometimes even a real console to test the various parameters for HTTP requests.

For the QueryLab portal, the communication through the various API services is hidden by a software layer that manages the setting of the different parameters, such as the http address of the services, registration keys and other data needed to obtain the answers to the queries requested by the user. The layer manages also the results, often provided in JSON (JavaScript object notation) format, which is simple to interpret and analyze with software tools because it provides both data and metadata at the same time. QueryLab translates the JSON format results into a graphical output to show the results.

After analyzing several API web services provided by museums and big data collectors around the world, we gathered a list of online repositories that have been successfully integrated into the QueryLab platform:

- **Europeana Collections**. It is the Europe digital platform for cultural heritage. Contributed by more than 3000 institutions across Europe, their assembled collections let users explore Europe's cultural and scientific heritage from prehistoric to the modern day (https://www.europeana.eu/en, accessed on 1 April 2022).
- **Victoria and Albert Museum**. The world's leading museum of art and design, it holds many of the UK's national collections and houses some of the greatest resources for the study of architecture, furniture, fashion, textiles, photography, sculpture, painting, jewelry, glass, ceramics, book arts, Asian art and design, theatre and performance (https://www.vam.ac.uk/, accessed on 1 April 2022).
- **Cooper Hewitt, Smithsonian Design Museum**. The only museum in the United States focused on historical and contemporary design, it is the curator of one of the most various and exhaustive design collections in existence—more than 210,000 design objects spanning 30 centuries (https://www.cooperhewitt.org/, accessed on 1 April 2022).
- **Réunion des Musées Nationaux—Grand Palais** collects the works of the most important French museums under the authority of the French Ministry of Culture (http://www.photo.rmn.fr/, accessed on 1 April 2022).
- **The Auckland War Memorial Museum** tells the story of New Zealand, its place in the Pacific and its people. The museum is a war memorial for the province of Auckland and has pre-eminent Maori and Pacific collections, significant natural history resources and major social and military history collections, as well as decorative arts and pictorial collections (https://www.aucklandmuseum.com/, accessed on 1 April 2022).
- **The Digital Public Library of America** contains materials from libraries, archives, museums and cultural institutions from all over America (https://dp.la/, accessed on 1 April 2022).
- **The National Museum of Australia**, the core collection of Australian history, collects and represents the aboriginal and Torres Strait Islander cultures and histories, Australian history and society since 1788 and people's interaction with the Australian environment (https://www.nma.gov.au/, accessed on 1 April 2022).
- **The Harvard Art Museums**, which include the Fogg Museum, Busch-Reisinger Museum, and Arthur M. Sackler Museum, are dedicated to advancing and supporting learning at Harvard University, in the local community, and around the world, playing a leading role in the development of art history, conservation, and conservation science (https://harvardartmuseums.org/, accessed on 1 April 2022).

## 4. A Three-Level Metadata Structure: ICH-Light Model

The analysis of the available data models for digital heritages made it clear that those available were suitable for tangible assets, such as objects, buildings, etc. The main problems encountered during the creation of the data model for ICH are related (a) to the different languages used for the content of the data, where English is often not among those available, making it difficult to understand the content, (b) to the lack of web services to access the data, (c) to the various categories of classification used that do not always follow those indicated by UNESCO, and finally (d) to the different methods used to catalog, preserve and disseminate the intangible cultural heritage.

Starting from these issues and the available data models, a novel data model designed on the specificities of intangible heritage is defined: we call it the ICH-light model [33].

The main goals of the model are to provide a framework that allow the following:

1. To develop content language-independent search tools;
2. To generate guided tours based on keywords, expressed in multiple languages;
3. To define an interchange format useful to realize a participatory system;
4. To improve the upload of contributions by the community.

Another important factor is the ease of mapping with the most suitable ontologies to transform the data into triplets and expose them for the LOD cloud.

Using a bottom-up approach, the model definition is started from the analyses of different inventories selected from the map of e-inventories [1]:

- ACCU Data Bank. The Asia Pacific Database on Intangible Cultural Heritage data was available only by html flat pages made by experts from the Asia-Pacific region, including Australia, Cambodia, Fiji, Kyrgyzstan, Tajikistan, Tonga and Vanuatu. Now the platform is no longer accessible but can be reached through WayBackMachine. Meanwhile, data have been saved into the model developed for QueryLab, acting also as safeguarding storage.
- Sahapedia is an open online resource on the arts, cultures, and heritage of India.
- The German Nationwide Inventory of Intangible Cultural Heritage.
- IntangibleSearch, the online collection of "living good" of Lombardy Region and Alp territories that are expressed through oral traditions, languages, performing arts, technical knowledge, social practices, rituals and festive events.
- Living Traditions in Switzerland, is the list of Swiss living traditions, recently updated.
- The data from the five inventories analyzed so far have generated the QueryLab IntangibleHeritage inventory, based on the ICH-Light model. The QueryLab portal integrates also other two cultural heritage local sources, to which we have access.
- LDA, Lombardy Digital Archive. This is the Archive of Ethnography and Social History of Lombardy Region that, since 1972, has preserved, studied and enhanced documents and images related to life and social transformations, literature and oral history, material culture, and the anthropic landscapes of the Lombard territory (http://www.aess.regione.lombardia.it/ricerca, accessed on 1 April 2022).
- MuFoCo, Museum of Contemporary Photography in C. Balsamo (Milan, Italy). It is a significant example of contemporary photographs and a cross section of photography after World War II to the present days and contains a subset of over one million and eight hundred thousand photographic works—prints in black and white and color images, slides, negatives, videos, installations—taken by about five hundred Italian and foreign authors. (http://www.mufoco.org, accessed on 1 April 2022).

The model's structuring phase requires incremental steps of reviewing and refinement, resulting in a data model organized into three layers to allow for different phases of data storage.

**Index Level**: the basic level dedicated to the main data of the identified ICH assets (Figure 2). It contains the minimal set of information expressed in a preferred language: English, or the native one when English is not available. The information at this level concerns the source archive, one or more domains of relevance, the place where the asset

lives or is handed down by communities, the name and a brief description, the link to the original resource and, when possible, some keywords, and finally a representative image. Keywords are processed through a glossary to associate, if possible, a translation in different languages (Italian, English, French and German so far).

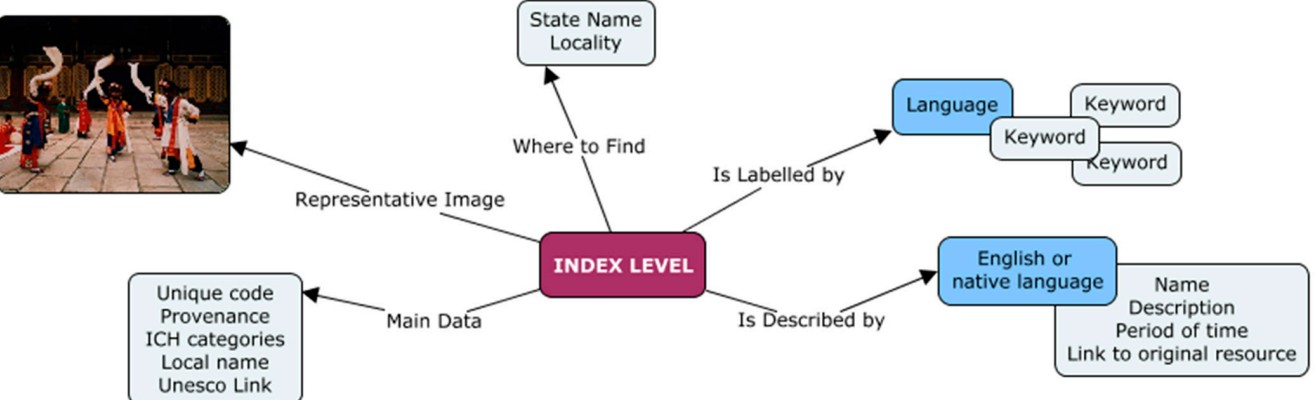

**Figure 2.** Index level.

**Catalog Level**: designed to enrich and integrate the index level, it allows to detail the catalogued assets and enables data entry in different languages, with respect to information on communities, valorization, and safeguarding measures. It is a useful model for inventory data not coming from a database but from simple descriptive pages, allowing a complete cataloging of the objects.

**Instantiation Level**: dedicated to the collection of the different instances of cataloged ICH assets, which include data and multimedia content useful to deepen and witness the evolution and adaptation of the ICH entity over time.

As shown in Figure 3, the instantiation layer contains the physical objects that describe the cataloged ICH entity, organized by date and place of detection.

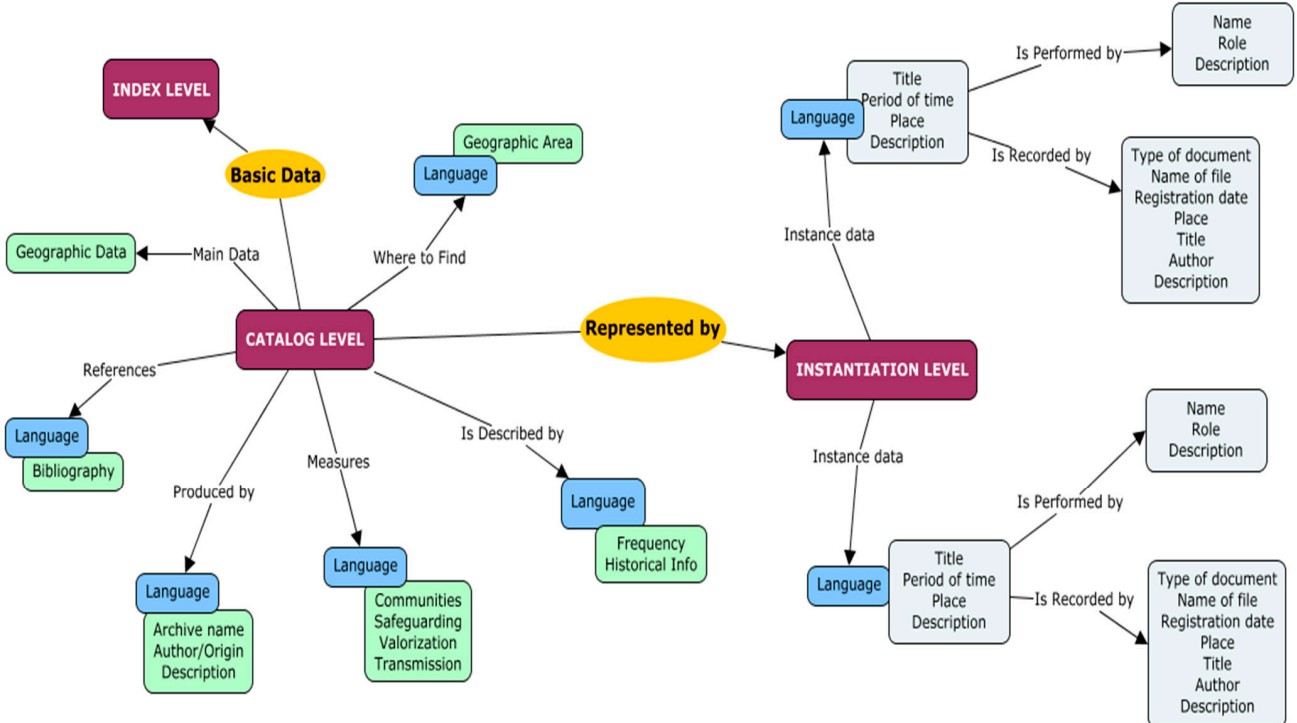

**Figure 3.** Catalog and instantiation level.

The model was tested on the five archives described above, inserting the data in the index level in a manual or semi-automatic way by ad hoc procedures. The data structured in this way allowed the use of search tools that abstracted from the different languages in which the contents are expressed and, through the analysis of the associated keywords, granted the generation of guided tours, which are language independent. QueryLab's ability to search the data at index level provided an initial qualitative assessment of the validity of the defined data model. Moreover, the index level is web-services ready and facilitates the creation of a participatory system that allows people of the communities to interact with the system and contribute to the shared index.

## 5. An Ontology Definition for ICH-Light Model

The definition of the data model is the first step toward the construction of a standard for the management of intangible assets data on the web. The next step requires the mapping of all concepts on an ontology: the preference is to maximize the reuse of what already exists and to create/define what does not find a correct mapping for the semantics of the data.

The modeling phase is based on the work of Szekely et al. [12], which uses the Europeana data model (EDM) [34] as a basis, and on the CHDE model (Cultural Heritage in Digital Environments) [13,14]. The concept of instantiation of the CHDE model (Cultural Heritage in Digital Environments) as well as the one-to-one principle of metadata [15] are integrated into the ICH-light model. The instantiation concept provides a contextual and historical description of the ICH entity; a set of different instances create the link between the various documentation collected during representations performed by different people in different times. When represented using a timeline, they can witness the living good changes and adaptations over time.

The ICH-light model's main object can be modeled by the class edm: Europeana Aggregation, used to describe the EDM instance, formed by edm: Provided CHO for the main description and by the class instantiation (to be created as new), needed to describe the digital resources of the ICH asset. The two classes are referred by the edm: aggregated CHO and edm: hasView properties (Figure 4).

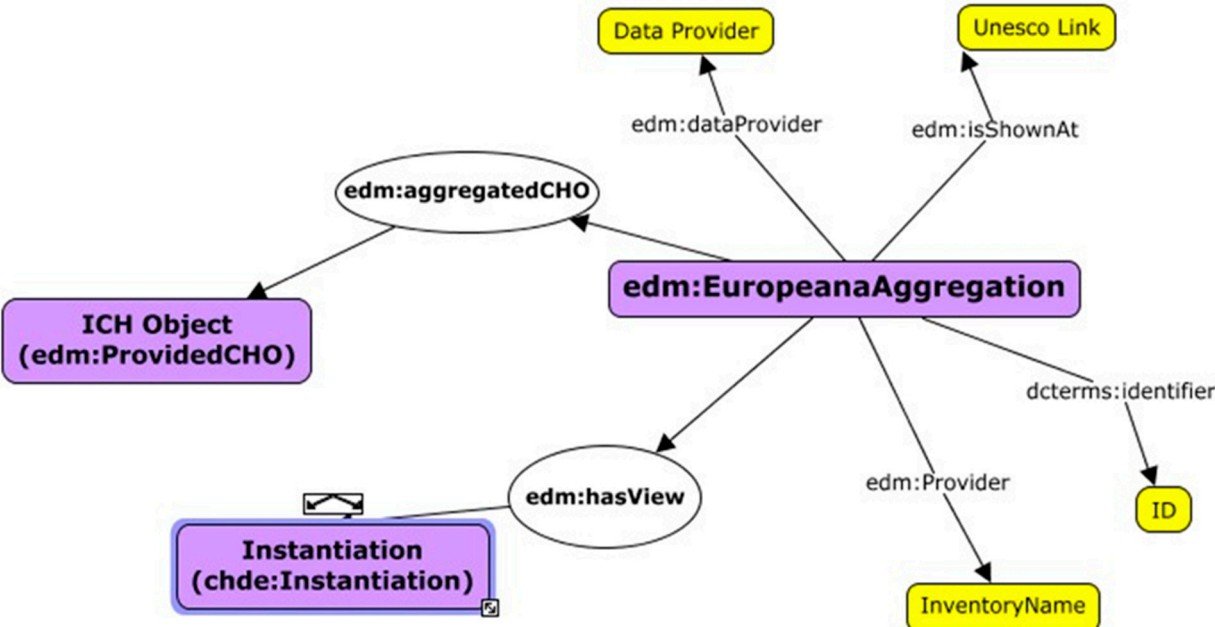

**Figure 4.** Main EDM instances.

All the classes and properties matched with the EDM model are shown in Figures 5 and 6, the new properties to be defined are represented by the red color: is-described-by, community-actions, instantiation-of, aggregated, is performed by, has skill, etc.

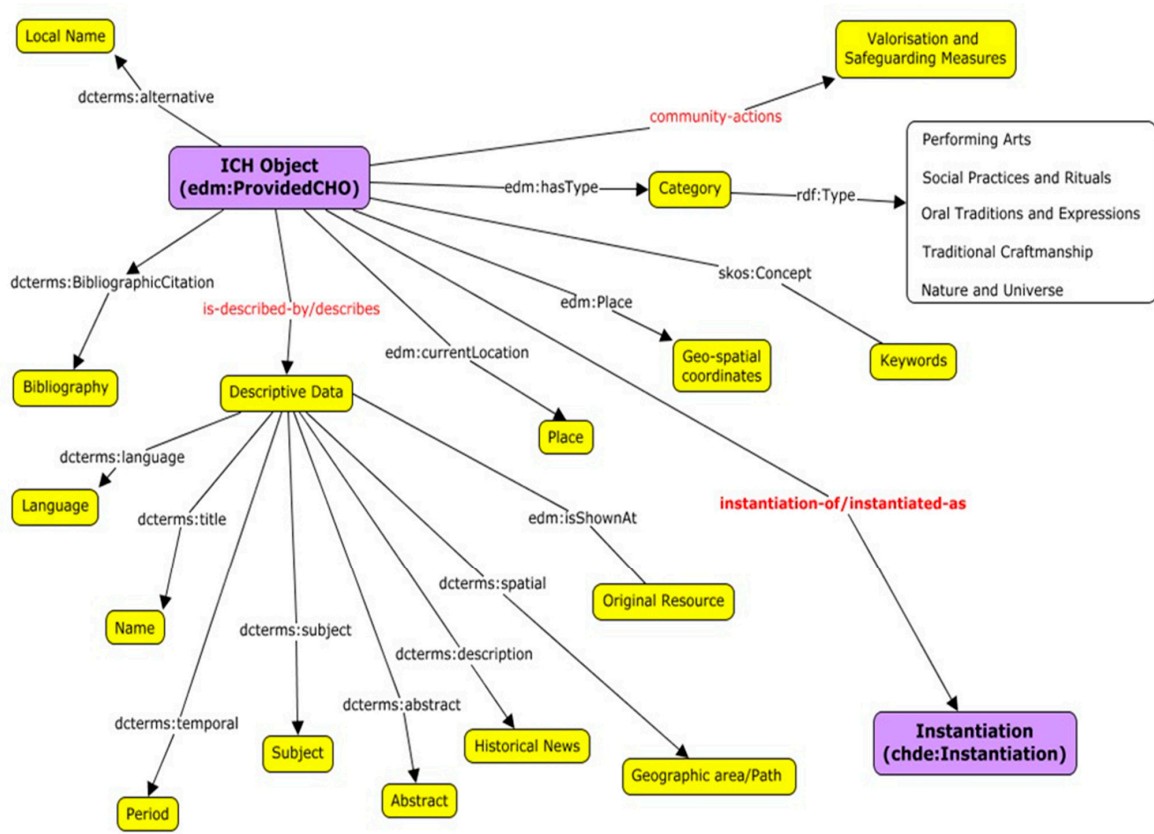

**Figure 5.** ICH object classes.

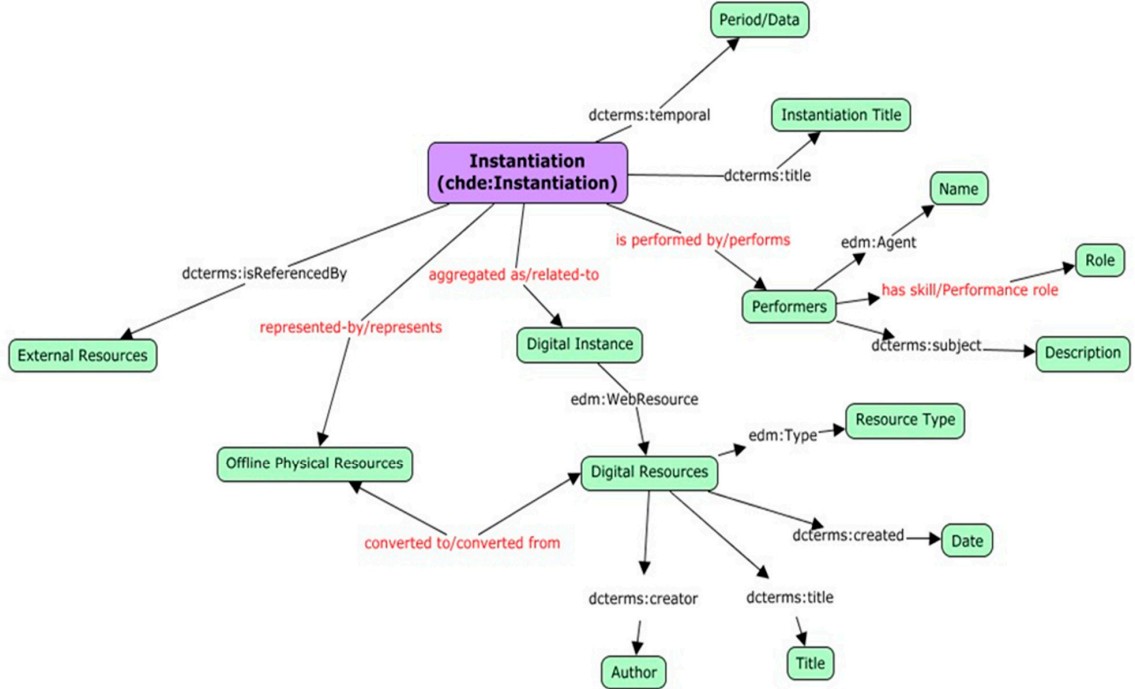

**Figure 6.** Instantiation classes.

## 6. Multimodal Search Engine

The QueryLab platform defines and implements a multimodal search engine, to increase user satisfaction in searching, browsing, viewing, and interacting with data from different sources. The goal is to allow different types of users to interact with the website comfortably and successfully, even if they are not experts in the field, or unfamiliar with the content or the language in which the terms are worded.

With the idea that the user should not be left with an empty text box as the only mode of interaction, coming from our experience in designing and building data-driven websites, several search and navigation tools have been defined and implemented, presented here in ascending order of automation. The possibility to specify search keys in the classic text box is present in QueryLab anyway, in order to offer the easiest and fastest way to query the repository to users who know perfectly what to search for or come back to the site to update or deepen their search.

### 6.1. Themed Routes

The first level of interaction implemented is the simplest exploration tool that requires no knowledge of data but provides satisfactory results. The "predefined queries" created in QueryLab are built by combining the work of ethnographers and experts with the glossary of keywords translated into different languages. The experts identified the semantic tags and organized them hierarchically on different levels, while the association with the different translations of the glossary allows searching regardless of the content language of the various archives. QueryLab offers users a user-friendly graphical interface to interact with and, on the other hand, experts have straightforward tools to easily update and create new predefined queries. Additional tags can be inserted and organized at any time to highlight relevant themes and topics introduced by the different inventories added over time.

Themed routes are offered with graphical presentation organized into three levels, and users can explore and retrieve documents simply by clicking on each suggested topic. The query is propagated directly into all the inventories involved. Figure 7 shows how keywords are organized into a tree used to generate the graphic interface, and Figure 8 shows the results for the category "Rituals/Traditional dance" performed in Europeana and Harvard Art Museum, where different types of objects can be found: images together with musical transcriptions, audio, and video.

### 6.2. Semantic Query Expansion

QueryLab offers a free text search mode according to the user's information needs. To ensure data and document retrieval, QueryLab integrates query expansion (QE) tools [26]. Here, the user's initial query is reformulated by adding other meaningful terms with similar meaning or translated into other languages.

QueryLab provides tools for expanding search terms, enlarging or reducing results, and suggesting hypernyms, hyponyms, or related terms (siblings of the same concept), by the integration of MultiWordNet [35]. WordNet is a large lexical database in English, where nouns, verbs, adjectives, and adverbs are grouped into cognitive synonym groups (synsets), each expressing a distinct concept. MultiWordNet is a multilingual version of WordNet that contains translations in several languages, such as Italian, Spanish, Portuguese, etc.

By MultiWordNet interaction, it is possible to suggest, in a semi-automatic way, the terms to be used to expand the query. The user can perform more specific queries, identifying the correct hyponyms: it can be particularly useful when querying databases such as DPLA with millions of documents. On the other hand, the generalization of a term or concept can be helpful in the case of small archives or if the user is interested in something particular: in this case, the query expansion with hypernyms can lead to greater user satisfaction.

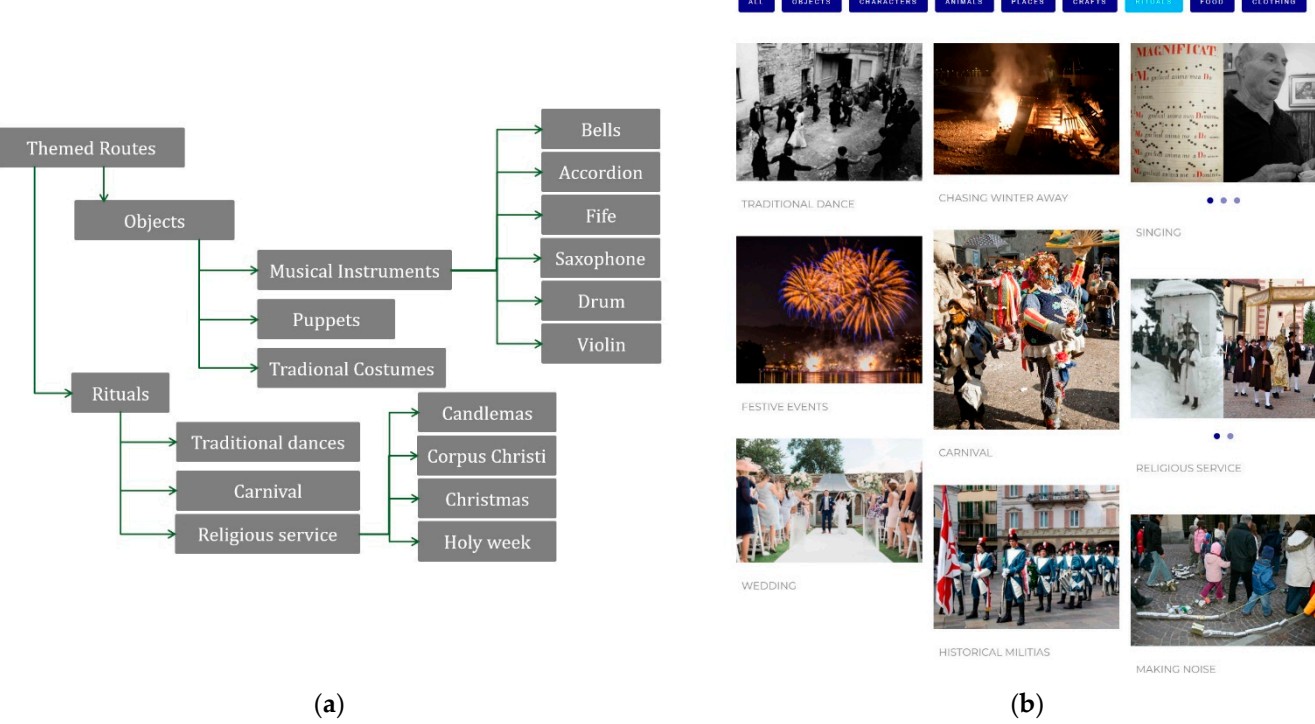

(**a**)                                                              (**b**)

**Figure 7.** Themed routes: (**a**) keywords structured by experts; (**b**) the theme "Rituals" with graphic presentation.

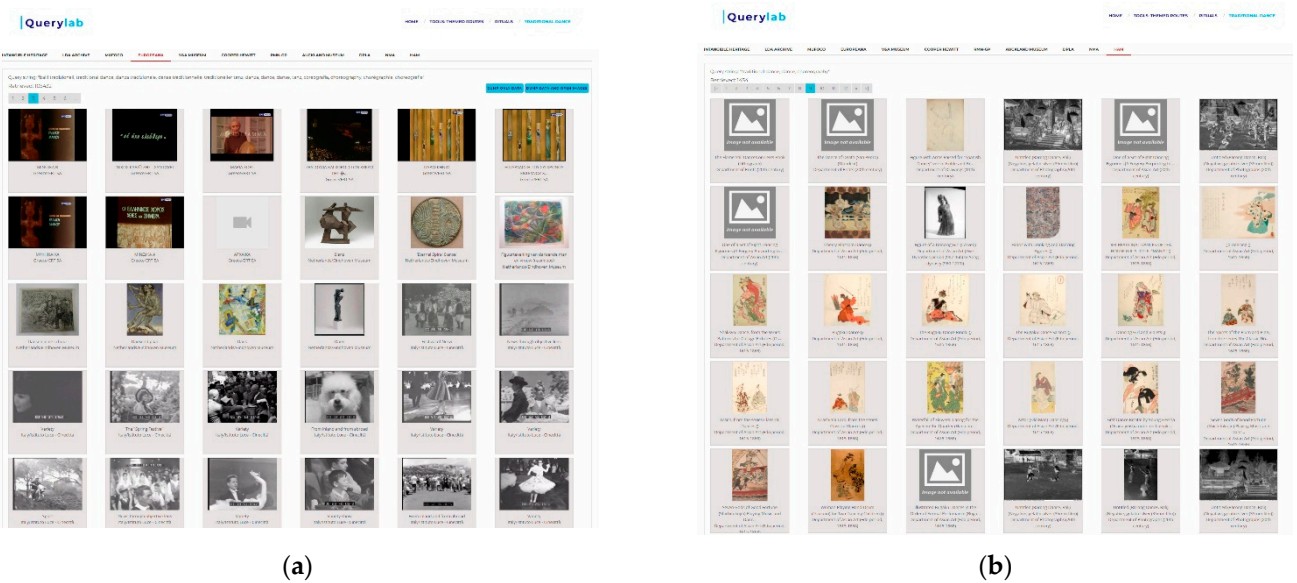

(**a**)                                                              (**b**)

**Figure 8.** Results for query "Rituals/Traditional dances": (**a**) from Europeana archives; (**b**) from Harvard Art Museum.

Moreover, MultiWordNet integration [36] allows translating terms into any Multi-WordNet language and structuring a flat list of terms into a tree glossary.

Figure 9 shows all coordinated terms of the selected tag "wedding rituals" and the results obtained by selecting one specific synset from the list: all terms of the synset, together with available translations, are used for searching.

These QE tools are used on local database tags. For web inventories, visual suggestions are proposed as described below.

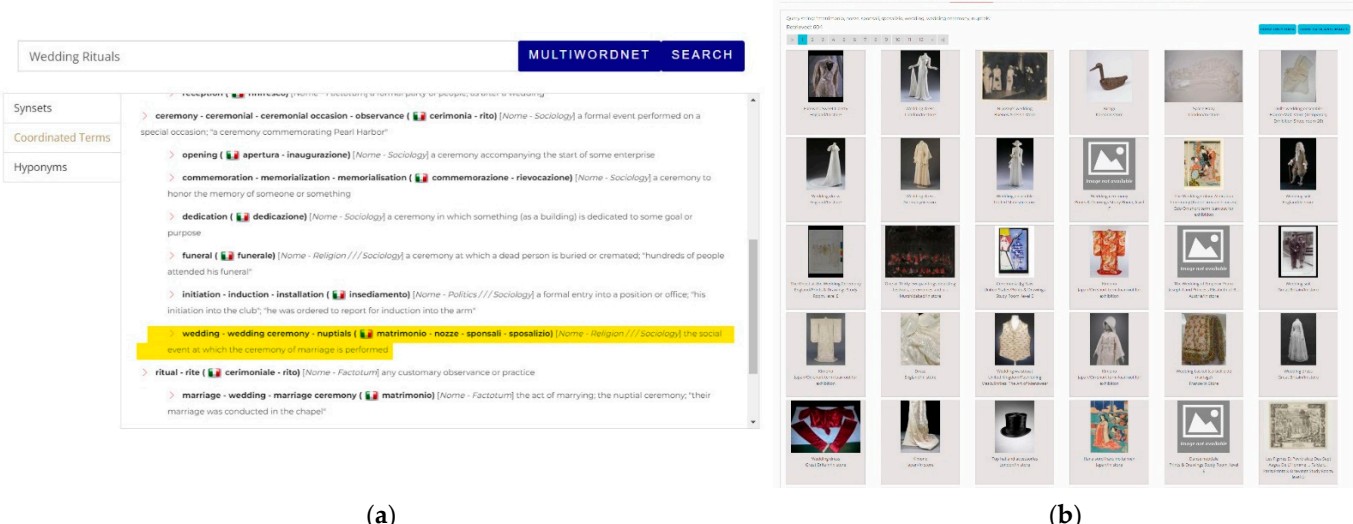

<div align="center">

(**a**)  (**b**)

</div>

**Figure 9.** Tag "Wedding Rituals": (**a**) MultiWordNet synsets; (**b**) results for the selected synset from the Victoria and Albert Museum.

### 6.3. Visual Suggestions

Web inventories are queried using the RESTful protocol: their data—including tags—are not collected locally, so the actually used tags in the web archives cannot be presented to the user in a clickable list.

One of the inspiring principles of QueryLab is to always offer the user at least an indication of the contents of the archives and to give suggestions about possible queries. We are currently studying a search engine that extracts, for each query made by the user, the most relevant tags associated to the retrieved documents, and then displays them as word clouds [37]. These word clouds offer new search cues in the current database.

The creation of the word list that forms the basis of the visual suggestions is a multi-step process:

1. Based on the documents retrieved by a query—simple, QE or thematic path—both tags and all terms are extracted, excluding stop words. The visual suggestion engine executes additional RESTful queries, customized for each QueryLab web repository, for the extraction of the data from the top n items. Different n values are tested, with n = 100, 200, 500.
2. The list of tags or terms in the title, description, etc. is created, sorted by occurrences. Each inventory returns different descriptive fields, according to the internal structure.
3. A word cloud is created for the tags and another for terms in the descriptive data.
4. Clicking on a tag or term, a new query is performed, which propagates to all available databases, and the process is repeated.

Even though the length of the titles/descriptions is quite short, ways of extracting only the significant words are being studied, by integrating algorithms of keyword extraction [38] or by weighing the terms according to TF-IDF weighting or information retrieval borrowed schemas [39].

The purpose of the visual suggestion is to provide a snapshot of the content of the different archives so as not to leave the user with the difficulty of an empty search box. Moreover, if the results obtained are not satisfactory, because they are too few and too insignificant, the word clouds can help in suggesting synonyms or alternative terms to use in the new query.

Of the remote databases in QueryLab, the Réunion des Musée Nationaux (RMN-GP) is currently the only one that makes use of a language other than English. Therefore, the query must be made in the language of the database—French—by translating the terms,

using MultiWordNet or the guided paths. In this case, visual suggestions are even more important because the tags or terms are extracted in French, constituting a sure help for users with little or no knowledge of the language.

The visual suggestion tool offers a simple and direct way to use tags/terms in the archives, improving the user experience and, with its simplicity and its ability to extract tags in the language of the archive (and not necessarily in English), it offers one more tool to improve the user's capability to retrieve the objects of interest, in any language.

Figure 10 shows the tag word clouds for the Digital Public Library of America (DPLA, USA), Réunion des Musée Nationaux (RMN, France) and Victoria and Albert Museum (England), respectively, for the query 'wedding'. For RMN, it was necessary to make the query in French ('mariage' in French). Word clouds were constructed on the frequency of tags in the first 500 results for each archive. The integration of this search mode on the tags allows easy use and the confidence of obtaining results.

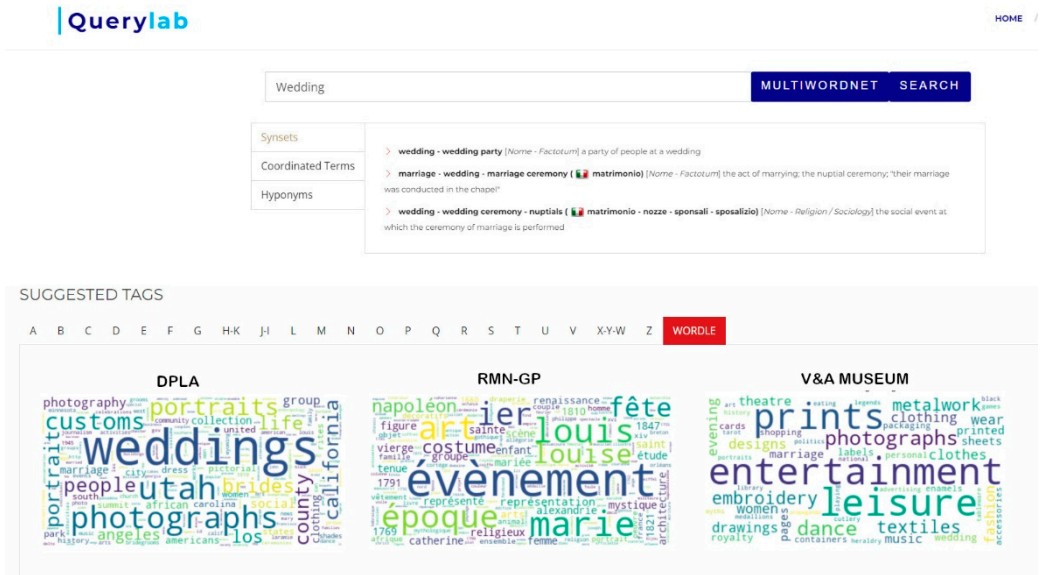

**Figure 10.** Tag word cloud for Digital Public Library of America (DPLA, USA), Réunion des Musée Nationaux (RMN, France) and Victoria&Albert Museum (England), respectively, for the query 'wedding' ('mariage' in French).

### 6.4. ICH Discovery

The ICH discovery section is dedicated to an innovative visualization of the QueryLab IntangibleHeritage inventory, which, thanks to multi-step refinement queries, enable the best use of metadata available at index level [40].

Users have several filters available to perform queries, which can also be combined with the themed routes available.

At first, all the data are presented through a simple mosaic of images, on the left, the available filters, and on the top right, the methods of displaying data, alternative to the mosaic, which give a graphically clearer idea of the distribution of the results. Users are allowed to make and refine queries, crossing values organized into the accordion filters. Results can be displayed using icon view or the graphs implemented, switching from one to the other.

The graphs experimented and developed till now, with different views, are as follows:

**Sunburst** is a three-level interactive circle, available in three different views, highlighting the country, domain or inventory source. Each section can be navigated by clicking on it, while the ICH assets, in the outer level, can be opened to show details;

**Force Graph** shows ICH assets linked to their domain, country or inventory, combined in different ways according to the chosen view. The graph is interactive because objects can

be dragged and dropped to investigate their links, and each ICH asset, represented by its image, can be selected and opened to view details by the ctrl key on the keyboard.

**Wind Rose** highlights the distributions of the number of ICH assets with respect to a couple of keys: the first is represented with the colors of the legend, while the second is represented on the circle line.

**Bubbles**, like the wind rose, shows the distributions of objects found as bubbles that are grouped together in a container, which can be the country, the category, or the inventory they belong to, according to the view selected.

**Dependency Wheel** displays the query keys on the outer circle, related to the different views available, and they are connected by strings which have a thickness commensurate with the number of objects meeting the connected criteria.

**Sankey**, like the dependency wheel but drawn on a plane, better highlights the connections of the key in the center with the other two on the left and right. The key in the middle is related to the different views available.

Figure 11 shows all the 1097 objects collected by the QueryLab IntangibleHeritage so far, displayed by sunburst/country on the left and sunburst/domain on the right. The main key is displayed in the inner circle, and for each key, in the next circle, the actual values are computed, while in the outer circle, we find all the ICH assets grouped by key/value. The main keys available are country, domain or source; by selecting on them, the graph is redrawn. This representation gives immediate feedback on the type of distribution of the query results about the countries, sources and categories.

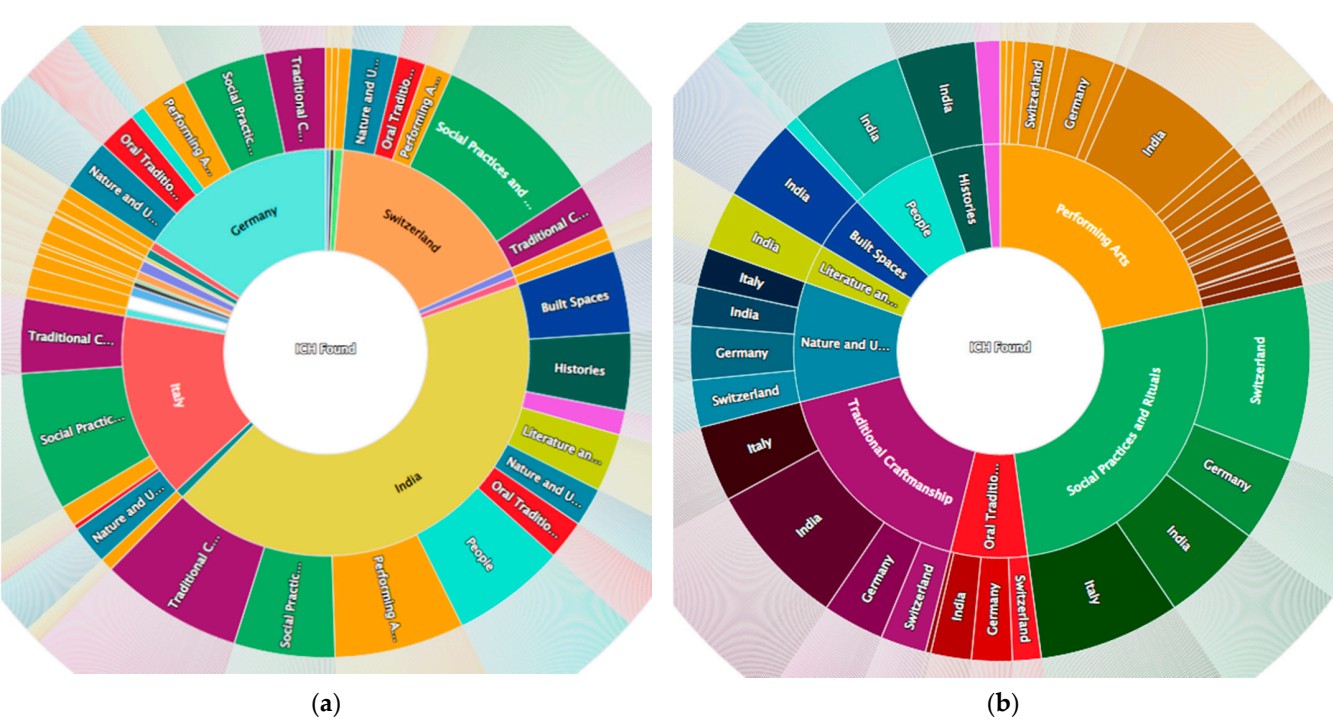

(**a**)  (**b**)

**Figure 11.** All ICH items displayed by sunburst: (**a**) the main key for the inner circle is the country; (**b**) the main key is the domain.

The sunburst graph is interactive and can be navigated by clicking on each graph section to better detail the next levels. Figure 12 on the left shows the results from the query "Theme = Traditional dances", while on the right, it shows ICH items refined by selecting "India" from the inner circle.

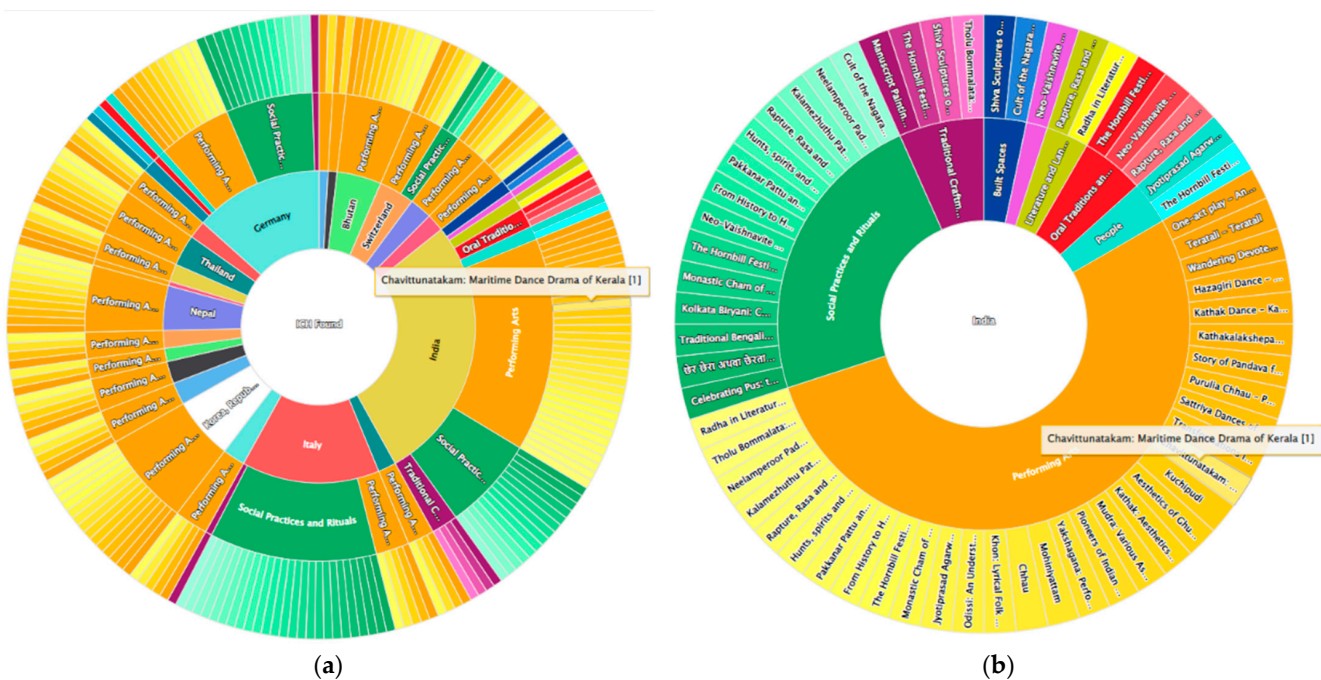

**Figure 12.** Results for theme "Traditional dances" by sunburst: (**a**) main key for the inner circle is the country; (**b**) selection of key "India" from the inner circle.

Figure 13 shows the ICH items resulting from the selection of the theme "Traditional dances", and the refinement with the "India" key on the right, represented by the force-graph. This graph is associated with a legend, where the color is connected to the corresponding domain, used also to draw connections between the domain and country or sources. The countries are identified by their flags, while for inventory sources, we used the available logo.

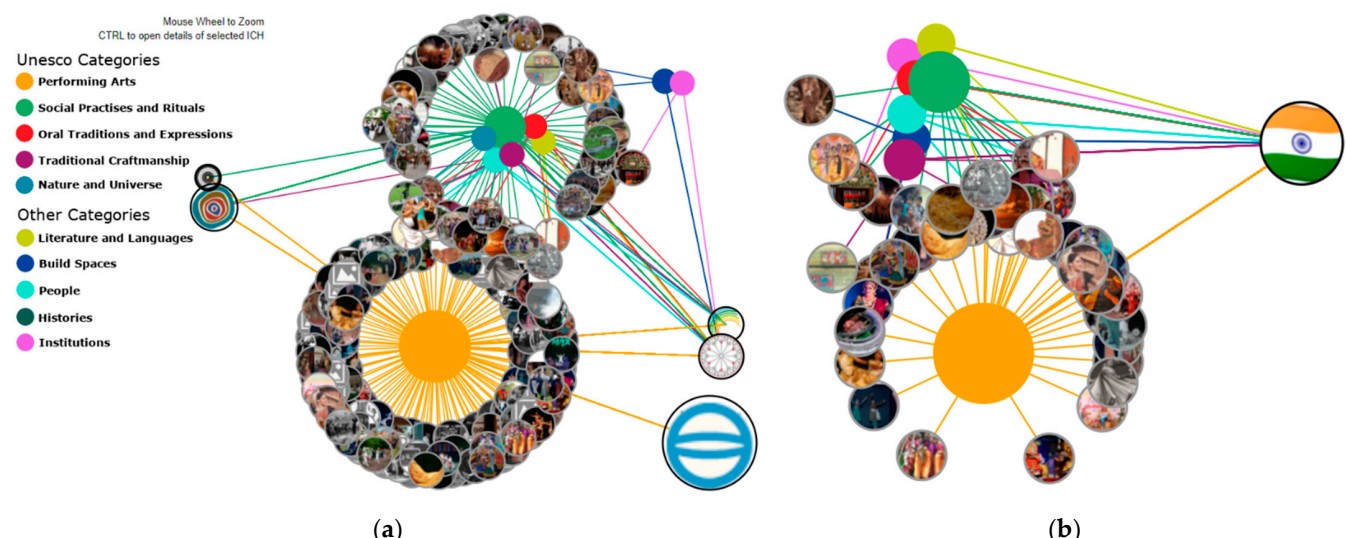

**Figure 13.** Results for theme "Traditional dances" by force graph: (**a**) view selected is "Source-Domain-ICH"; (**b**) refinement with the key "India" from filters.

Figure 14 shows the same results displayed by the dependency wheel and Sankey graph. Both connect the keys involved, two by two, with strings that have a thickness proportionate to the amount of ICH assets that satisfy the key pair conditions. The color used is relative to the domain whenever possible.

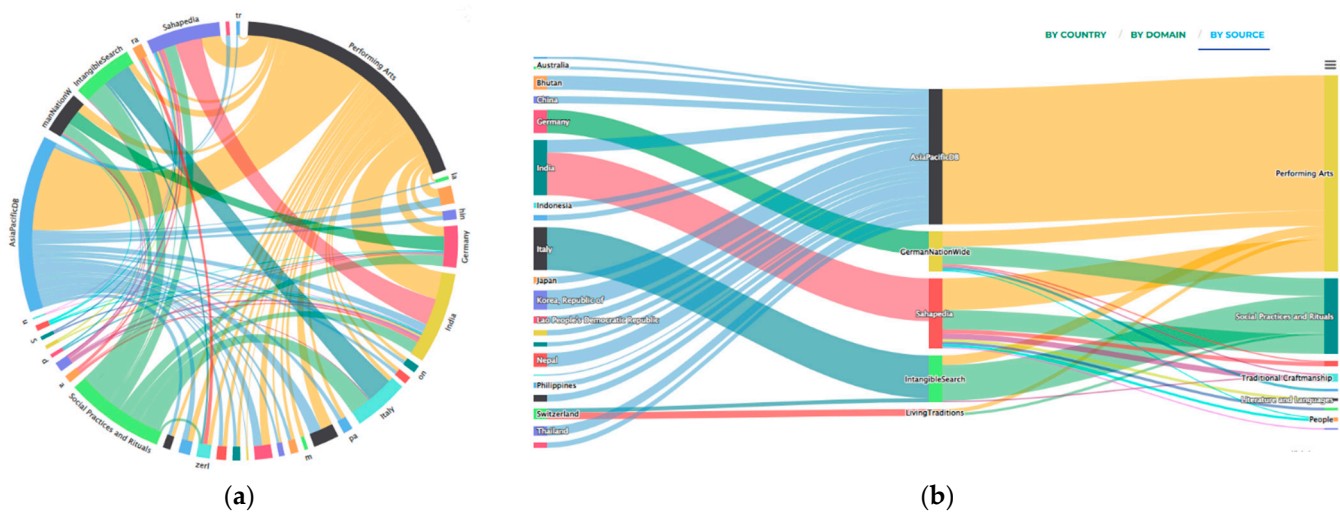

(**a**)　　　　　　　　　　(**b**)

**Figure 14.** Results for theme "Traditional dances": (**a**) dependency wheel graph; (**b**) Sankey graph.

Figure 15 shows all ICH items displayed by the bubbles and wind rose graphs. The bubbles graph is shown by the source inventory view, while wind rose shows the number of objects for the couple domain/country: the domain key is on the circle, while countries are represented by colors indicated by the legend on the right. Each graph can be switched using the different views available: domain, country or inventory source.

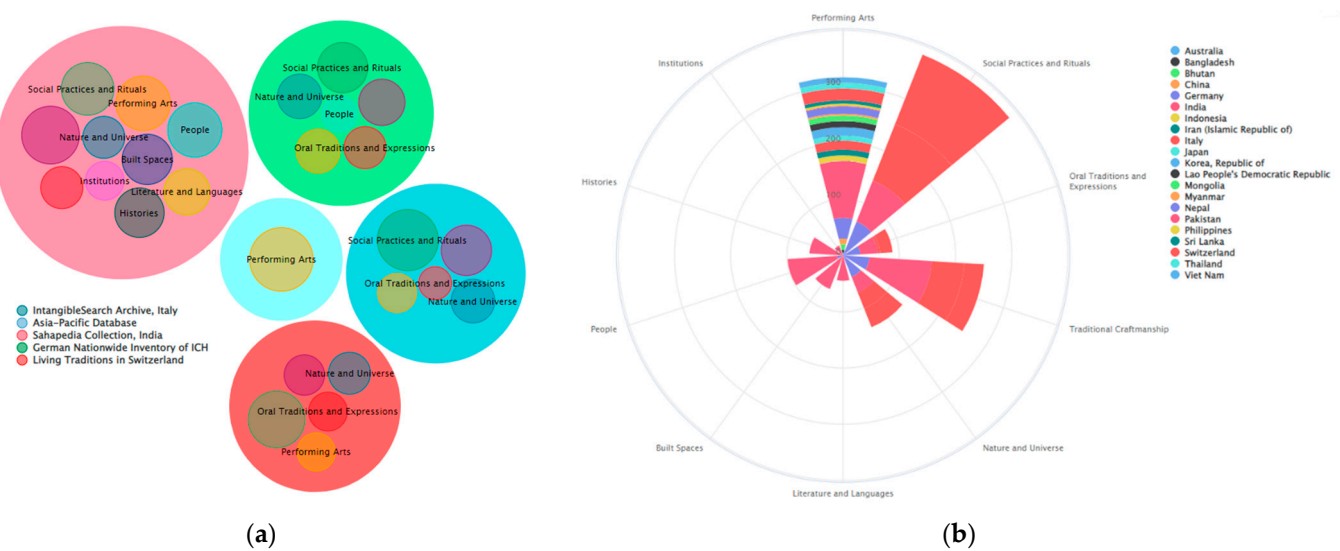

(**a**)　　　　　　　　　　(**b**)

**Figure 15.** All ICH items displayed: (**a**) bubble graph; (**b**) wind rose graph.

## 7. Discussion

The QueryLab platform has been completely designed, defined, and implemented. Data, both local and web queried, are continuously evolving and growing.

QueryLab's different ways of integrating inventories allow to collect and manage data in a transparent way for the user. Now, there are eight web archives, with a constantly growing heritage and are immediately made available on QueryLab. The number of items ranges from over fifty-one million Europeana items to over two hundred thousand of the Cooper-Hewitt, National Museum of Australia and Harvard Art Museums; more than 500,000 photographs of artwork housed in France's national and regional museums addressed by RMN-GP; 900,000 records and 300,000 openly licensed images from Auckland Museum; more than 1.2 million objects from Victoria and Albert Museum; and 1097 to date for the ICH-light Archive, related to five inventories.

QueryLab was conceived to integrate, in the most automatic way possible, archives both via web services and local ones. As far as new web inventories are concerned, integration requires updating the software layer (see Figure 1): after a phase of analysis of the characteristics of the specific API interface, a dedicated query procedure is added to the layer, customizing query parameters, paging and return methods, as well as the access key. The time required to update is relatively short, and the procedure semi-automatic. What needs to be evaluated is the actual ability to extract subsets of data, as web services sometimes provide simple extractions of the entire catalog, without the query capability. As for adding a new 'silos' inventory to the QueryLab Intangible Heritage, the feasibility of the scraping algorithm must be evaluated on a case-by-case basis. It is possible to automate the process when the web pages have a homogeneous structure, otherwise it must be done manually. The most complex part concerns the identification of the web pages related to the inventory contents to be ingested, which can rarely be performed in a completely automatic way. Integration times vary depending on the possibility of automation.

The ICH-light model for indexing different archives dedicated to intangible heritage has resulted in a structure organized on several levels to guarantee a flexible and customizable organization that facilitates community participation. The experimentation on different inventories, five have been identified so far, has allowed a first qualitative evaluation of the created model. The data of the simplest level of the structure allow both to store useful information and to allow efficient and innovative queries, as shown in the portal and in the section ICH Discovery specifically.

Finally, a first mapping with the classes and properties of ontologies specific to this area was proposed. Those classes and properties, specific to the model, not present in any ontology with the semantic specification were added.

The search and fruition tools, designed for different types of users, allow searching by keywords or texts, for those who know what to look for, or through more advanced tools, such as thematic paths and word clouds: it is possible to retrieve information from many inventories at the same time, even when the content of the inventories is in a different language from one's own. The query expansion allows users to widen or narrow the search field, using consolidated tools, such as WordNet. Preliminary results have shown that the new QueryLab tools are easy to use and improve search efficiency by visualizing results from different archives in one scenario and in different forms, facilitating the sharing, dissemination, and preservation of cultural heritage.

To minimize user waiting time, when a user executes a search, the system queries one archive at a time, the first presented is the IntangibleHeritage inventory, based on the ICH-light model, and prepares the queries for the other archives. Only when the user clicks on the corresponding tab is the query actually executed.

The response times—in general, very fast—vary from archive to archive and depend on (i) whether the archive is local or remote; (ii) the size of the archive queried; and (iii) the number of items retrieved. Obviously, the speed of the network and the busyness of the remote archive influence the response time. The type of search also affects the speed. For example, when you make an expansion query with MultiWordNet, the system prepares a query with hyponyms, hypernyms or siblings: this affects performance because of the complex queries.

Development was designed for multiple devices, focusing on mobile systems, using Bootstrap 4.0, JavaScript and the PHP language for development, along with MYSQL for the underlying database. The use of REST API web services to query the inventories allows updating with additional archives at any time.

## 8. Conclusions and Future Works

The paper presented QueryLab, a prototype dedicated to the integration, navigation, searching and preservation of tangible and intangible heritage archives on the web, with the aid of themed paths, keywords, semantic query expansion and word cloud. Results can be grouped and shown using alternative ways to the classic mosaic of images, and

the original page of each document retrieved can be reached immediately. Starting from the research and analysis of archives on tangible and intangible assets, together with the study of metadata standards for cultural heritage, in Italy and worldwide, in this article, we presented a novel metadata structure, articulated in three levels, for the specific archiving of intangible assets. A proposal for a mapping of the data model into an ontology was also presented: the ontology is still in the definition phase, and once completed we intend to perform tests, comparisons with what is already available, and to offer details on it. We also intend to evaluate if and how the ontology allows to increase QueryLab content at the index level by integrating data exposed as LOD.

The platform was fully designed and some features were implemented and tested with users. Preliminary evaluations showed that the system is appreciated by users.

Some issues emerged in the search interface for the expansion query. At the moment, all available tags for local archives are proposed as a starting point: this causes a preponderance of Sahapedia-specific tags. We intend to experiment with extracting keywords or tags from remote archives as well to make more terms available, thus decreasing the importance of terms used only by one archive. We want to mimic the information retrieval methods of introducing weights to give more importance to words that appear not too often and not in all archives, such as TF-IDF or similar, on the tags/terms extracted by the visual suggestion engine.

Future developments will also cover the following:

- Creation of "near me" and "calendar" features, to allow the exploration of ICH assets around the user in place and time;
- Creation of faceted indexes, to help in refining queries from inventories reached by web services;
- Introduction of automatic keyword extraction algorithms to automatically structure thematic paths;
- The creation of general word clouds for web archives, both on the tags and on the words of the title/description, in addition to those linked to specific queries.
- The creation of a markup schema useful to improve the way search engines read and represent ICH assets pages, to be presented as a standard on schema.org [41,42].

In addition, we would like to understand which of the search tools best responds to the needs of different types of users, through the administration of a questionnaire.

**Author Contributions:** Conceptualization, M.T.A. and I.G.; metadata modeling, M.T.A.; software, M.T.A. (mainly), I.G.; writing—original draft preparation, M.T.A., I.G.; writing—review and editing, M.T.A. All authors have read and agreed to the published version of the manuscript.

**Funding:** This research received no external funding.

**Institutional Review Board Statement:** Not applicable.

**Informed Consent Statement:** Not applicable.

**Data Availability Statement:** Data used for supporting reports come from the inventories detailed in Sections 3 and 4.

**Conflicts of Interest:** The authors declare no conflict of interest.

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
