# Peer review of "Integrating, Indexing and Querying the Tangible and Intangible Cultural Heritage Available Online: The QueryLab Portal"

_information, doi:10.3390/info13050260_

Round 1

Reviewer 1 Report

Dear Authors,

thanks for allowing me to review your manuscript. The topic is fascinating and the QueryLab prototype is really ineresting. That's said, this work describes this prototype but fails to link it this paper to other studies. What is the theoretical background for this study? What are the theoretical implications? Was your aim to present this prototype or do you discuss its role from an academic point of you. If the aim was to present the QueryLab, very interesting as I've said, then make it clear in the article type and in the introduction as well. Currently, the introduction does not contain any scientific references.

Reviewer 2 Report

The paper is a summary article on the overall workings of the QueryLab system. While the involved domains digital humanities and semantic data integration are a perfect fit, the actual contribution of this paper is not well described nor made explicit.

Nearly every aspect of the paper - including the pictures without explicitly referencing! - besides mabe the second part of Chapter 4 with the instance/class figures have been published before. If chapter 4 should be the major contribution you have to include much more details on the number of concepts, relations, classes, ... some evaluation regarding scalability, especially if you actually really want to use some form of inference later on!

In short, I would make it clear from the beginnng that this is a summary article and contribute performance/scalability aspects such as query times, etc. (missing in all your other articles as well mentioned in your paper). Additionally, include the coverage of your processes regarding manually, semi-automatic and automatic as well as the time needed to get a feeeling of applicability!

Reviewer 3 Report

This manuscript describes the development of the QueryLab portal that is designed to query multiple cultural heritage databases through an integrated interface. Technology-wise, the novelty of this project is limited. However, it is quite an endeavor to build such a system that can facilitate the researchers in the domain of cultural heritage.

Round 2

Reviewer 1 Report

Dear Authors,

this version seems to be another manuscript, the changes are really numerous. You left the comments in Italian in the version that you uploaded. Is this the final version you wanted to submit?

Overall, the argumentation is clear and the manuscript has been significantly improved. Please check the link between the paragraphs.

Reviewer 2 Report

The authors addressed the feedback of the reviewers well and clarified self-plagiarism concerns.

The paper is now a well structured and presented broad overview article of the QueryLab system highlightning specialities and challenges.
